# Molecular Epidemiology of SARS-CoV-2: The Dominant Role of Arginine in Mutations and Infectivity

**DOI:** 10.3390/v15020309

**Published:** 2023-01-22

**Authors:** Harry Ridgway, Charalampos Ntallis, Christos T. Chasapis, Konstantinos Kelaidonis, Minos-Timotheos Matsoukas, Panagiotis Plotas, Vasso Apostolopoulos, Graham Moore, Sotirios Tsiodras, Dimitrios Paraskevis, Thomas Mavromoustakos, John M. Matsoukas

**Affiliations:** 1Institute for Sustainable Industries and Liveable Cities, Victoria University, Melbourne 8001, VIC, Australia; 2AquaMem Consultants, Rodeo, NM 88056, USA; 3Institute of Chemical Biology, National Hellenic Research Foundation, 11635 Athens, Greece; 4NewDrug PC, Patras Science Park, 26504 Patras, Greece; 5Department of Biomedical Engineering, University of West Attica, Egaleo, 12210 Athens, Greece; 6Laboratory of Primary Health Care, School of Health Rehabilitation Sciences, University of Patras, 26504 Patras, Greece; 7Institute for Health and Sport, Victoria University, Melbourne 3030, VIC, Australia; 8Immunology Program, Australian Institute for Musculoskeletal Science (AIMSS), Melbourne 3021, VIC, Australia; 9Pepmetics Inc., 772 Murphy Place, Victoria, BC V6Y 3H4, Canada; 10Department of Physiology and Pharmacology, Cumming School of Medicine, University of Calgary, Calgary, AB T2N 1N4, Canada; 114th Department of Internal Medicine, School of Medicine, National and Kapodistrian University of Athens, 11527 Athens, Greece; 12Department of Hygiene Epidemiology and Medical Statistics, School of Medicine, National and Kapodistrian University of Athens, 11527 Athens, Greece; 13Department of Chemistry, National and Kapodistrian University of Athens, 11571 Athens, Greece; 14Department of Chemistry, University of Patras, 26504 Patras, Greece

**Keywords:** SARS-CoV-2, COVID-19, epidemiology, mutations, ACE2, ARBs, RAS, RBD, arginine, proteases, infectivity

## Abstract

**Background**, **Aims**, **Methods**, **Results**, **Conclusions:** Severe acute respiratory syndrome coronavirus 2 (SARS-CoV-2) is a global challenge due to its ability to mutate into variants that spread more rapidly than the wild-type virus. The molecular biology of this virus has been extensively studied and computational methods applied are an example paradigm for novel antiviral drug therapies. The rapid evolution of SARS-CoV-2 in the human population is driven, in part, by mutations in the receptor-binding domain (RBD) of the spike (S-) protein, some of which enable tighter binding to angiotensin-converting enzyme (ACE2). More stable RBD-ACE2 association is coupled with accelerated hydrolysis by proteases, such as furin, trypsin, and the Transmembrane Serine Protease 2 (TMPRSS2) that augment infection rates, while inhibition of the 3-chymotrypsin-like protease (3CL^pro^) can prevent the viral replication. Additionally, non-RBD and non-interfacial mutations may assist the S-protein in adopting thermodynamically favorable conformations for stronger binding. This study aimed to report variant distribution of SARS-CoV-2 across European Union (EU)/European Economic Area (EEA) countries and relate mutations with the driving forces that trigger infections. Variants’ distribution data for SARS-CoV-2 across EU/EEA countries were mined from the European Centre for Disease Prevention and Control (ECDC) based on the sequence or genotyping data that are deposited in the Global Science Initiative for providing genomic data (GISAID) and The European Surveillance System (TESSy) databases. Docking studies performed with AutoDock VINA revealed stabilizing interactions of putative antiviral drugs, e.g., selected anionic imidazole biphenyl tetrazoles, with the ACE2 receptor in the RBD-ACE2 complex. The driving forces of key mutations for Alpha, Beta, Gamma, Delta, Epsilon, Kappa, Lambda, and Omicron variants, which stabilize the RBD-ACE2 complex, were investigated by computational approaches. Arginine is the critical amino acid in the polybasic furin cleavage sites S1/S2 (681-PRRARS-686) S2′ (814-KRS-816). Critical mutations into arginine residues that were found in the delta variant (L452R, P681R) and may be responsible for the increased transmissibility and morbidity are also present in two widely spreading omicron variants, named BA.4.6 and BQ.1, where mutation R346T in the S-protein potentially contributes to neutralization escape. Arginine binders, such as Angiotensin Receptor Blockers (ARBs), could be a class of novel drugs for treating COVID-19.

## 1. Introduction

Unraveling the molecular mechanisms of mutations of SARS-CoV-2 is the key to the discovery and design of new treatment targets for the Corona Virus Disease 2019 (COVID-19). In this commentary, we report (1) the epidemiology distribution of variants across European countries and variant prevalence (dominant earlier alpha, beta, gamma, and delta variants now are replaced by omicron variants [1,2,3,4]) and (2) dominating mutations N501Y, E484K, K417N, P681H, P681R, and D614G, which are attempted to be explained based on computational Molecular Dynamics (MD) simulations and on structured traits of the residues in their charged configuration in the interface between SARS-CoV-2 and ACE2 before and after developing mutations [5,6,7,8,9,10,11]. The trend of the mutations is the replacement of non-polar and hydrophobic residues with polar and hydrophilic amino acids, which are able to create more stable ligand-binding networks, allowing the mutant to increasingly spread [12]. Intermolecular pi–pi interactions observed between Y41 of the ACE2 receptor and various aromatic substitutions at the N501 locus of the RBD indicated that such interactions enhanced the RBD-ACE2 binding [12]. The order of aromatic mutations enhancing binding was tryptophan (W) > tyrosine (Y) > phenylalanine (F). Aromatic interactions potentially play an essential role in increasing binding [13,14,15,16,17]. The notion that more polar residues favor stronger RBD-ACE2 binding is supported within the context of this sub-trend since tryptophan and tyrosine residues are more polar than phenylalanine (the most lipophilic amino acid). Furthermore, tryptophan with an excess of aromatic pi electrons compared to tyrosine and phenylalanine results in stronger binding with ACE2. Tryptophan and tyrosine also participate in hydrogen bonds (HB) with ACE2 residues through the heteroatom nitrogen of the W ring and the hydroxyl group of the Y ring (not possible for the side chain of the F residue), which is consistent with the polarity ranking. Comparison of the wild-type and the mutant protein stabilities by free energy calculations, as well as calculations of protein–protein intermolecular interactions’ free energies, are in accordance with enhanced RBD-ACE2 binding predicted by residue polarity. The structure of the SARS-CoV-2 S-protein’s RBD bound to the ACE2 receptor and mutations that strengthen SARS-CoV-2 infectivity have been reported [5,12,18]. This article focuses on the molecular epidemiology of SARS-CoV-2 and the driving forces that trigger mutations in SARS-CoV-2 [13,14], where arginine seems to play a dominant role.

## 2. Materials and Methods

The ECDC provides variants’ distribution data regarding the 30 EU/EEA countries on a weekly basis. The distribution is based on the sequence or genotyping data regarding the SARS-CoV-2 detection and variant classification that are deposited in the GISAID and TESSy databases. Table 1 contains the summary data regarding the variants overview reported on 24 November 2022 (46th week of 2022), as mined from the official ECDC’s website ([19] accessed on 24 November 2022). Only variants that are considered as variants of concern (VOC) or variants of interest (VOI) (as of 24 November 2022) are included in the summary depicted in Table 1 (all omicron subvariants). The weekly variant distribution plot between weeks 34–45 of 2022 is also provided by ECDC, based on data from previous weeks.

Docking of ligands to the ACE2 receptor in the RBD-ACE2 complex was performed using AutoDock VINA14 as implemented in the Yet Another Scientific Artificial Reality Application (YASARA) software suite. Global docking of ligands to the ACE2 receptor in the RBD-ACE2 complex (PDB: 6LZG) was performed using AutoDock VINA14 [20] using default parameters. Partial atomic charges and dihedral barriers were initially assigned according to the AMBER03 force field [21] and then damped to mimic the less polar Gasteiger charges used to optimize the AutoDock scoring function. Energy terms are based on the ECEPP/3 force field [22,23], an all-atom vacuum force field with appended terms for solvation-free energy and entropic contributions. The setup was conducted with the YASARA molecular modeling program [24]; the best hit of 900 runs per ligand was reported as kcal/mol free energy of binding.

Molecular dynamics (MD) simulations were performed with the YASARA suite [25]. The setup included definition of periodic boundaries, a pKa prediction to fine-tune the protonation states of protein residues at the chosen pH of 7.4 [26], and optimization of the hydrogen bonding network [27] to increase the solute (i.e., receptor or receptor–ligand) stability. Sodium and chloride ions were introduced to a physiological concentration of 0.9 wt%, with an excess of either Na^+^ or Cl^−^ to neutralize the cell. After the steepest descent and simulated annealing minimizations to remove clashes, simulations were typically run for a minimum of 40 nanoseconds (ns) using the AMBER14 force field [28] for the solute, GAFF2 [29] and AM1BCC [30] for ligands, and TIP3P for water. The cutoff was 8 Angstroms (Å) for Van der Waals forces (the default used by AMBER [31]), and no cutoff was applied to electrostatic forces (using the Particle Mesh Ewald algorithm [32]). The equations of motions were integrated with a multiple timestep of 2.5 femtoseconds (fs) for bonded intramolecular interactions and 5.0 fs for non-bonded interactions at a temperature of 311 K and a pressure of 1 atm (NPT ensemble) using algorithms described in detail previously [33].

## 3. Results

### 3.1. Variant Distribution across EU/EEA Countries

In the past three years, the pandemic caused by COVID-19 was accompanied by the emergence of new variants of SARS-CoV-2 [34]. The five dominant VOCs that have been reported so far (as of November 2022) are alpha (B.1.1.7), beta (B.1.351), gamma (P.1), delta (B.1.617.2), and omicron (B.1.1.529) ([35], accessed on 24 November 2022). The first two variants (i.e., alpha and beta) were responsible for the peak of COVID-19 infections and deaths from October 2020 to the beginning of summer 2021 (depending on the area), while the third variant (i.e., gamma) was mainly circulated during the same period in Brazil and L. America [36]. The third wave of COVID-19 was caused by the fourth variant, named delta, which circulated from April 2021 until the end of the year. The fifth variant was named omicron and was responsible for the next wave of COVID-19 infections, achieving its highest rate around the globe in December 2021 and January 2022. The omicron variant was initially characterized for its vaccine breakthrough and immune evasion [18], and as such, it affected the epidemiological situation in the following months. The initial omicron variant (named BA.1) was soon replaced by the omicron BA.2 subvariant in March 2022, which was subsequently and rapidly replaced by two new subvariants, named BA.4 and BA.5, in April 2022. All omicron sub-variants have been reported to possess at least 50 mutations in their genome and are the most mutated variants containing 31–37 mutations in the S-protein compared to the previous variants of concern [1].

In the current phase of the pandemic, the infections caused by the BA.5 subvariant remain the most dominant across COVID-19 patients in the European continent ([19] accessed on 24 December 2022). Interestingly though, two sublineages named BA.2.75 (“Centaurus”) and BQ.1 (“Cerberus”) can also be detected within the EU/EEA countries in accountable percentages, and with the latter to be gaining ground according to the latest increasing rates of infections. The BQ.1 subvariant, as of 20 October 2022, is considered a VOI from ECDC ([37], accessed on 24 December 2022). BA.2.75, a sublineage variant of BA.2, on 7 July 2022, was added as a variant under monitoring (VUM) by ECDC and, as of 24 December 2022, is considered as VOI. BQ.1 originates from the BA.5 omicron’s VOC. BQ.1.1 is another sublineage of the original BA.5 variant and exhibits high potential to replace BA.5 and become the new dominant variant. Based on modeling estimates, by the beginning of 2023, more than 80% of COVID-19 cases are expected to be due to BQ.1/BQ.1.1 SARS-CoV-2 variants [18,36,37].

Sequence or genotyping data regarding the SARS-CoV-2 detection and variant classification through the GISAID and the TESSy databases are summarized and reported on ECDC’s official website on a weekly basis [19]. Table 1 contains the summary data regarding the variants overview reported on 24 November 2022 (46th week of 2022). An adequate weekly sequencing volume is a level at which it is possible to follow trends and estimate the proportion with sufficient precision (variant’s prevalence of 5% or lower). Higher numbers of sequences increase the accuracy and allow the detection of variants accounting for a smaller proportion of circulating viruses. As can be seen in Table 1, the volume of sequencing or genotyping for variant detection can be classified into four categories: level 1a (L1a), level 1b (L1b), level 1c (L1c), and level 2 (L2). L1a is for sequence or genotyping volumes capable of variant proportion estimations with sufficient precision at the variant prevalence of 1% or lower, L1b for >1–2.5%, L1c for >2.5–5%, and L2 for >5%.

ECDC provides variants’ distribution information regarding the 30 EU/EEA countries (Table 1). For the period between weeks 44 and 45 of 2022 (produced on 24 November 2022), 24 out of the 30 countries report adequate sequence or genotyping volumes (L1a, L1b, L1c, or L2 levels). For the six countries that have not reported sequence or genotyping data during that period, this category is annotated as “No data” (i.e., Croatia, Cyprus, Finland, Lithuania, Malta, and Slovakia). Only variants that are considered as VOC or VOI (as of 24 November 2022) are included in the summary depicted in Table 1 (all omicron subvariants). These variants are BA.5, BQ.1, BA.2.75, BA.4, and BA.2. Any de-escalated VOC or VOI or variants reported as ‘Other’.

**Table 1 viruses-15-00309-t001:** Overview report regarding the SARS-CoV-2 variant distribution for EU/EEA countries ([19] accessed on 24 November 2022).

Country	Weeks of Data ^1^	Data Source ^2,3^	Number of Cases	Volume of Sequencing or Genotyping for Variant Detection			Total Known Variants Detected	BA.5		BQ.1		BA.2.75		BA.4		BA.2		Other	
				*n*	%	Category		*n*	%	*n*	%	*n*	%	*n*	%	*n*	%	*n*	%
Austria	44–45	GISAID	54,210	4272	7.9	L1a	740	410	55.4	99	13.4	138	18.6	32	4.3	5	0.7	56	7.6
Belgium	44–45	GISAID	8953	235	2.6	L2	235	122	51.9	91	38.7	7	3	2	0.9	5	2.1	8	3.4
Bulgaria	44	TESSy	2893	166	5.7	L2	166	165	99.4	1	0.6								
Croatia		GISAID	0	0	0	No data	0												
Cyprus		TESSy	0	0	0	No data	0												
Czechia	44–45	GISAID	12,317	53	0.4	L2	53	48	90.6	4	7.5			1	1.9				
Denmark	44–45	TESSy	8450	5629	66.6	L1a	5629	3023	53.7	1852	32.9	560	9.9	57	1	137	2.4		
Estonia	44–45	TESSy	1043	608	58.3	L1b	608	579	95.2			11	1.8	5	0.8	12	2	1	0.2
Finland		GISAID	0	0	0	No data	0												
France	44–45	GISAID	305,759	866	0.3	L1c	861	394	45.8	410	47.6	8	0.9	3	0.3	32	3.7	14	1.6
Germany	44–45	TESSy	443,282	4661	1.1	L1a	4661	4275	91.7			151	3.2	182	3.9	53	1.1		
Greece	44–45	TESSy	87,883	387	0.4	L2	387	271	70	40	10.3	42	10.9	8	2.1	4	1	22	5.7
Hungary	45	TESSy	4431	168	3.8	L2	168	165	98.2					1	0.6	2	1.2		
Iceland	44–45	GISAID	615	145	23.6	L2	145	51	35.2	62	42.8	19	13.1	7	4.8			6	4.1
Ireland	44–45	TESSy	3315	188	5.7	L2	188	71	37.8	94	50	10	5.3	4	2.1	5	2.7	4	2.1
Italy	44–45	GISAID	298,878	1300	0.4	L1b	1290	835	64.7	373	28.9	27	2.1	12	0.9	22	1.7	21	1.6
Latvia	44–45	TESSy	5415	1086	20.1	L1b	1086	1072	98.7			5	0.5	5	0.5	3	0.3	1	0.1
Liechtenstein	44–45	GISAID	139	24	17.3	L2	24	8	33.3	7	29.2	9	37.5						
Lithuania		GISAID	0	0	0	No data	0												
Luxembourg	44–45	TESSy	2809	667	23.7	L1c	667	319	47.8	276	41.4			1	0.1	17	2.5	54	8.1
Malta		TESSy	0	0	0	No data	0												
The Netherlands	44	TESSy	8226	701	8.5	L1b	701	430	61.3	186	26.5	46	6.6	7	1	20	2.9	12	1.7
Norway	44–45	TESSy	1786	105	5.9	L2	105	64	61	29	27.6	8	7.6	1	1	3	2.9		
Poland	44–45	GISAID	5964	59	1	L2	59	57	96.6	1	1.7	1	1.7						
Portugal	44–45	TESSy	11,310	242	2.1	L2	242	225	93					14	5.8			3	1.2
Romania	44–45	TESSy	4995	189	3.8	L2	189	154	81.5	1	0.5	3	1.6	2	1.1	1	0.5		
Slovakia		TESSy	0	0	0	No data	0												
Slovenia	44	GISAID	4565	9	0.2	L2	9	8	88.9	1	11.1								
Spain	44–45	GISAID	37,601	377	1	L2	374	128	34.2	225	60.2	17	4.5	1	0.3			3	0.8
Sweden	44–45	GISAID	7151	756	10.6	L1c	756	461	61	233	30.8	19	2.5	4	0.5	14	1.9	25	3.3

^1^ Data produced on 24 November 2022 (week 46). ^2^ GISAID database: [38]. ^3^ TESSy database [39], accessed on 24 November 2022.

The European variants’ landscape evolves around the BA.5 variant prevalence (Figure 1). The BA.2, BA.2.75, and BA.4 variants’ proportions are de-escalating. The BQ.1 variant is ranked 2nd for 16 out of the 24 countries that reported data for weeks 44–45. The BA.2.75 variant is ranked 2nd for 7 out of the 24 countries (i.e., Austria, Estonia, Germany, Greece, Latvia, Liechtenstein, and Romania). Poland reports equal percentages (i.e., 1.7%) for both the BQ.1 and the BA.2.75 variants. Interestingly, the BQ.1 variant is gaining ground, as it exhibits increasing proportions in the last 6 weeks’ period (from week 39 to week 45 of 2022) for most of the EU/EEA countries, as can be seen in Figure 1.

### 3.2. Arginine Mutations Stabilize Best the Conformation of S-Protein and the Complex with ACE2

The role of arginine in replacing the wild-type RBD residues in critical positions 452, 484, and 501, where mutations happen, is dual. One is to stabilize the conformation of the S-protein’s conformation, which facilitates the molecular recognition with ACE2, and the second is to stabilize the RBD-ACE2 complex further. Docking studies have shown higher stability for both the conformation of S-protein and the RBD-ACE2 complex after selective arginine mutations [13,14]. The chemical structure of arginine (Figure 2A) indicates that the strong positive charge of the guanidino group is shared through resonance with the three nitrogen atoms, rendering this group a super binder with anionic groups [40]. This is shown in docking interaction studies between bisartan A (4 butyl imidazole bearing two N,N′ biphenyl tetrazole groups) and ACE2 (PDB: 6LZG) catalytic center residues. One tetrazole group of bisartan A is interacting with the guanidino groups of ACE2 arginine residues, Arg518 and Arg514, and the second tetrazole is interacting with the guanidino groups of Arg518 and Arg273 (Figure 2B). The ability of arginine to bind stronger compared to other amino acids is also seen in stability and docking studies of dominant mutation RBD 501R with ACE2 anionic residues [12]. The guanidino group of mutation 501R binds strongly through salt bridges and pi–pi interactions with acidic (E and D) or aromatic residues (W, Y, F, and H) of ACE2, as well as with the ionic moieties, tetrazoles, or carboxylates of angiotensin II receptor blockers (ARBs) [13,14]. The chemical structure of BisA is seen in Figure 2C.

Our previous studies have shown that mutations at position 501, from asparagine to tyrosine and then to arginine, progressively increase the stability of the S-protein’s conformation due to strong interactions of arginine with neighboring residues. S-protein’s stability is expressed by a change in total free energy (ddG) or free energy of solubility (ddSol) as a function of the mutation type (amino acid substitution) at the 501 locus of RBD. Thus, an asparagine > tyrosine > arginine substitution resulted in the largest enhancement of protein stability with the ranking order to be as follows: ddG = 0 kcal/mol for the wild type (N501), ddG = −0.24 kcal/mol for the Y501 mutant, and ddG = −1.28 kcal/mol for the R501 mutant (lower ddG values translate into higher stability). The stability of the RBD-ACE2 complex is the most important factor regarding SARS-CoV-2 epidemiology [12]. The higher stability of the RBD-ACE2 complex results in higher infectivity and transmissibility. Intermolecular interactions of residues that hold together the proteins of RBD and ACE2 vary in strength and depend on the structural features (polarity, aromaticity) of the interacting residues [12]. These interactions are hydrophobic, hydrophilic, hydrogen bonding, salt bridges, and pi–pi interactions, which predominate. The crystal structure of the complex (PDB: 6LZG) has identified the wild-type RBD N501 interaction with ACE2 Y41 through hydrogen bonding which binds the two chains. Mutation N501Y, the dominant mutation in the alpha variant [6], increased the interaction with ACE2 Y41 through pi–pi interactions enhancing the stability of the complex. N501R mutation was shown to increase the interaction with ACE2 Y41 further [13].

In Figure 3, the S-protein’s stability is illustrated as a function of mutations occurred by amino acid substitution at position 501 of the RBD (upper panel). The conformation of the three mutants at the 501 locus (i.e., N501, Y501, and R501) is also depicted in Figure 3 (lower panel), along with the ddG values for each mutation type. The highest interaction occurred for the mutant R501 (Figure 3C) [13].

The docked pose of the chlorinated analog bisartan C (BisC, N,N′ bisbiphenyl tetrazole losartan, the chemical structure is shown in Figure 4A) to the interfacial region between the ACE2 receptor and the RBD of SARS-CoV-2 is depicted in Figure 4B (van der Waals surface and ribbon representation). This pose resulted from the global docking of BisC to the RBD (PDB: 6LZG) using AutoDock VINA. The docking domain comprised a cuboid cell with non-periodic (wall) boundaries 8 Å from any target atom. The BisC binding motif (Figure 4C) primarily involved pi–pi (red lines), pi–cation/salt bridges (blue lines), and hydrophobic (green lines) interactions. The interactive residues of SARS-CoV-2 RBD were Tyr505, Arg403, Phe456, and Tyr421, Tyr473 and Lys417, while those of ACE2 were Ala387, Arg393, Pro389, Val93, Lys26, Leu29, Asp30, His34, and Glu23. The intermolecular interactions of BisC with the RBD included pi–cation/salt bridges with Arg403 and Lys417, pi–pi interactions with Tyr505, Tyr421, Phe456, and Tyr473, and one hydrophobic interaction with Lys421. The binding of BisC to the ACE2 interfacial region was mainly dominated by a plethora of hydrophobic interactions (green lines) with Ala387, Pro389, Val93, Lys26, Leu29, Asp30, and Glu23, one pi–pi interaction with His34, as well as one pi/cation (salt bridge) interaction (blue lines) with Arg393. The MD simulation for the ACE2-RBD-BisC complex revealed that the bound BisC molecule was moderately stable and remained in the binding pocket (Figure 4D). The MD simulation was run for 32 ns. Figure 4D depicts the heavy atom (HA) RMSD trends for the ACE2 receptor (orange), bound RBD (gray), and bound BisC (yellow) over the 32 ns MD simulation of the ACE2-RBD-BisC complex at 311 K and the constant pressure of 1 atm (NPT ensemble). Inset images show BisC conformations at 15 ns, 22 ns, and 32 ns. While BisC was relatively stable over the course of the MD simulation, it was not ejected into the aqueous phase. Brief periods of sporadic instability were noted (e.g., ~12–17 ns).

## 4. Discussion

### 4.1. Variant Distribution across EU/EEA Countries

In the current phase of the pandemic, the infections caused by the omicron SARS-CoV-2 variants remain the most dominant across COVID-19 patients in the European continent ([19] accessed on 24 November 2022). The BA.5 subvariant prevalence can be seen for all EU/EEA countries that report sequence or genotyping data through the GISAID and TESSy databases (Figure 1, Table 1), while the BA.2, BA.2.75, and BA.4 variants’ proportions are de-escalating. Interestingly, the sublineages BA.2.75 and BQ.1 can be detected within the EU/EEA countries in accountable percentages. BA.2.75 is a sublineage variant of BA.2 and, as of 24 November 2022, is considered as VOI. The BQ.1 sublineage seems to be gaining ground according to the latest increasing rates of infections, and as of 20 October 2022, it is considered as a VOI from ECDC ([37], accessed on 24 November 2022). BQ.1 originates from the BA.5 omicron VOC. BQ.1.1 is another sublineage of the original BA.5 variant and exhibits high potential to become the new dominant variant.

### 4.2. Polybasic Cleavage Sites of the SARS-CoV-2 S-Protein and 3CL^pro^ Inhibition

The critical P681R mutation at the rich arginine furin cleavage site (FCS) 681–686 (Figure 5A) enhances cleavage and infectivity suggesting a critical role for basic arginine residues in SARS-CoV-2 for spreading infection. A second S2′ cleavage site is involved in promoting infusion initiation, and the cleavage occurs between R815-S816 by trypsin and trypsin-like proteases, such as the TMPRSS2 [42,43,44]. The latter may also cleave at the S1/S2 FCS. A recent study suggests that furin might also cleave between R815-S816 at the S2′ cleavage site, with an additional FCS located at the N terminal domain (NTD) of the S protein [42]. Trypsin enhances SARS-CoV-2 infection in cultured cells [45] and, along with other proteases produced in the lungs or small intestine, might boost viral replication in these organs, ultimately leading to severe tissue damage. Furin, trypsin, and TMPRSS2 might act synergistically in viral entry and infectivity, supporting the combination of furin, trypsin, and TMPRSS2 inhibitors as potent antiviral drugs [46]. Figure 5B shows the homotrimeric SARS-CoV-2 S-protein and the S1/S2 and S2′ FCSs. The polybasic cleavage sites of SARS-CoV-2, 680-SPRRARS-686, and 810-SKPSKRS-816 provide potential treatment targets [42,44,47,48,49]. Interaction of polybasic cleavage sites (arginine’s positively charged side chain) occurs with negatively charged ARBs (Figure 2). ARBs, namely sartans, bear negative charges (tetrazolate and carboxylate) and may target the positive cavity loops of the S-protein. These positively charged arginine residues block the entry of the virus through ACE2. This applies to all sartans which bear negative charges, inlcuding bisartans which consist of two tetrazole groups (e.g., BisA, Figure 2C) [13]. In silico studies suggest that ARBs either block ACE2, inactivating the entry of the virus, or prevent hydrolysis of the S-protein by furin, trypsin, and TMPR552 [14]. Arginine is the critical amino acid for cleavage, and inhibition by ARBs may be critical for developing novel antiviral treatments.

The viral genome, after hijacking the host ribosomes, gets translated into ~800 kDa large polypeptide (PP) chain. The newly generated PP chain is autoproteolytically cleaved by proteases such as 3CL^pro^. The latter is encoded by the viral genome in order to generate the non-structural proteins (NSPs) that are required for replication. 3CL^pro^ is the main protease of SARS-CoV-2, and as such, it plays a major role in viral replication. The PP chain is cleaved into 16 NSPs in total. The 11 NSPs are generated by the 3CL^pro^, rendering this protease one of the major targets for developing anti-SARS-CoV-2 drugs [50]. Bisartans have been found in our previous in silico studies to bind strongly to the 3CL^pro^ catalytic center and to be stable in MD simulations [13,14].

### 4.3. Critical Mutations

The objective of this study is to explain the stability, higher affinity, and infectivity of the SARS-CoV-2 mutants based on the charge interactions in the new network environment formed after the mutation. Computational chemistry has revealed and predicted stronger binding to ACE2 for certain mutations at positions 501, 417, and 484 of RBD. Mutations of SARS-CoV-2 have been identified in several countries [5,7,8,9,51,52]. Natural products and S-protein’s fragments have been investigated in silico as possible inhibitors [53,54,55]. The most known mutations reported so far are N501Y, E484K, K417N, P681H, P681R, and D614G, with the three first to be in the RBD interface with ACE2 [5,8,9,52]. The N501Y mutation appeared to be the most infective in the UK Alpha variant, which also includes the less infective K417N and E484K mutations. The P681H mutation is known as the Nigerian lineage and with the D614G mutation, located in the A chain of the S-protein, are bound to a lesser degree compared to the three RBD mutants E484K, K417N, N501Y [7,10].

#### 4.3.1. The Triple Mutation E484K, K417N, N501Y (UK Variant)

It has been reported that the combination of E484K, K417N, and N501Y mutations results in the highest degree of conformational alterations in the S protein RBD when bound to ACE2, compared to E484K, K417N or N501Y alone [5,7,8,9,51,52]. The triple mutation together in the S protein RBD domain has an additive effect and increases the affinity of RBD for ACE2 [13]. In particular, the E484K mutation switches the charge on the flexible loop region of RBD, leading to the formation of novel favorable contacts. These mutations were examined under the light of network interactions with ACE2 residues and vary according to the polarity and the charge of the interacting residues, which create a new, more stable environment where they are favorably accommodated. The major forces that dominate the interactions are derived from salt bridges, hydrogen bonds, and pi–pi interactions. Mutations point strongly to the convergent evolution of SARS-CoV-2 RBD structures to improve binding affinity to the ACE2 receptors, which subsequently makes them more stable and more infective. Strong inter- and intra-interactions in the new setting make mutants survive and spread. Pi–Pi interaction, as in mutant Y501RBD/Y41 ACE2, was found to be the strongest compared to hydrogen bonds based on phenol-phenol interactions [8,9,10].

The N501Y mutation, known as the UK mutation, was initially dominating globally. The fact that the N501Y mutation appeared independently in different geographical areas after its first appearance in the UK provides further indication of a possible advantageous mutational shift of higher infectivity compared to the original N501 variant. The N501Y mutation allows the virus to bind to the ACE2 receptor more tightly compared to N501. In the crystal structure of RBD/ACE2 [12,32], the wild type’s N501 residue of the RBD (PDB 6LZG) is interacting through hydrogen bonding (3.85 Å) with ACE2 Y41 (Pi–donor HB). Additionally, the side chain’s oxygen atom of Y41 forms one conventional HB (2.73 Å) with the side chain of T500 of the RBD (donor). Mutation N501Y (PDB 7EKG) presumably leads to a conventional HB (2.73 Å) between the hydroxyl group of 501Y (donor) and the oxygen atom from the carbonyl group of the backbone of Y41 (acceptor) of the ACE2. The latter is slightly more stable compared to the conventional hydrogen bond in wild type variant in terms of donor-acceptor distance (Å). Enhancement of the affinity is further enforced by the quadrupole ring-ring interaction (pi–pi T-shaped) between the two tyrosine residues (4.87 Å), which strengthens the binding (Figure 6). Tyrosine is known to interact in a charged network through the nucleophile tyrosinate, as in angiotensin II (AngII) receptor activation, and through hydrogen bonding, salt bridges, and quadrupole ring-ring interactions with aromatic residues. This results in stronger binding in the network of charges with subsequent favorable spreading and infectivity. This mutation and the broad tyrosine interactions in the mutant that stabilize the binding are energetically favorable compared to N501, with an amide functionality of reduced reactivity compared to the tyrosine (aromatic, hydroxyl) functionality [8].

The enhanced complex stability imparted by N501Y was consistent with a decrease in the ddGbind of −0.36 kcal/mol compared to that of the N501, as revealed by our previous study [13]. In contrast, the isolated single-point mutations E484K and K417N were by themselves destabilizing, with ddGbind values of +0.5049 and +0.9442 kcal/mol, respectively, suggesting a weakening of the RBD-ACE2 binding. However, in MD simulations, both mutants (i.e., E484K and K417N) exhibited improved binding to ACE2. Their combined effect on the triple mutant (i.e., E484K, K417N, and N501Y) was to significantly increase the RBD-ACE2 binding potential. The ddGbind was significantly increased when compared to the wild-type and to any of the three single-point mutations separately. Moreover, based on this study’s MD results, the singular pi–pi interaction of the two tyrosine residues between RDB N501Y and ACE2 Y41 is what really drives the binding. A weak HB between N501Y-RBD and Y41-ACE2 can be observed after 9 ns of MD simulation between ACE2 and the RBD (PDB 6LZG), but it is quite distant (>2.5 Å) compared to the other 7 HBs. The single pi–pi interaction that is preserved after 9 ns can potentially overwhelm that weak HB in terms of the interaction potential. Secondly, due largely to the pi–pi interaction, the N501Y interaction with Y41 of ACE2 is the closest atom interaction (~1.6 Å) of all interactions detected. This proximity association is most likely due to the pi–pi interaction and not to the weak HB, although the latter could indeed make a partial contribution by withdrawing electron density from the aromatic ring(s). The pi–pi interaction could assume slightly different conformations following repeated energy optimizations (e.g., T-form, Sandwich, and Parallel).

#### 4.3.2. The Triple Mutation L452R, E484Q, P681R (Indian Variant)

The sublineage B.1.617.1 (kappa) appeared in India and included mutations E484Q, L452R, and P681R with severe symptoms in the population. The mutation E484Q (ddG = −0.29 kcal/mol) results in the mutant 484Q, which is a much stronger binder compared with the mutant 484K (ddG = +0.02 kcal/mol) according to our MD simulations, which predicted the stronger affinity and infectivity of the 484Q mutant [13]. The major triple combination of L452R, E484Q, and P681R seen in the Indian variant suggests that the virus is evolving similar traits Independently and continuously adapting to its human hosts. The replacement of hydrophobic amino acids L, E, and P with the hydrophilic R, Q, and R also suggests that the trend of virus is to reduce hydrophobic interaction and increase hydrophilic interactions with ACE2 residues which are stronger, creating more stable networks allowing the virus to survive, evolve, and spread [5,52].

#### 4.3.3. The E484K Mutation (UK, South Africa, and Brazil Variants)

Molecular dynamics simulations reveal that the E484K mutation, which appeared in UK, South Africa, and Brazil variants ([57], accessed on 24 November 2022), enhances S protein RBD-ACE2 affinity [52]. The combination of E484K, K417N, and N501Y mutations induces conformational changes greater than the N501Y mutant alone, potentially resulting in an escape mutant [7,51,52]. The increased affinity of 484K, compared to 484E, for ACE2, has been suggested to be due, in part, to the change in net charge. This allows the formation of a transient contact ion pair with E75 of ACE2 [8,9,12]. In particular, the strong salt bridge between the lysine amino group at position 484 of the S-protein and with a glutamic acid carboxyl group at position 75 of ACE2 increases the affinity of RBD for ACE2 [12].

#### 4.3.4. Comparative Analysis of the E484Q and E484K Mutations

The E484Q mutation, which appeared in the kappa variant (B.1.617.1), resulted in an escape mutant where glutamate in position 484 was replaced by glutamine, an amino acid of a similar hydrophilicity index [58] (Table 2). The mutant 484Q was highly infective, and the S-protein’s RBD possessed a higher affinity towards the ACE2 residues, as predicted and also confirmed in our previous MD simulation studies [13,14]. Structurally, glutamine contains a carboxamide (-CONH_2_) which can interact with ACE2 residues through carbonyl and amino groups capable of forming hydrogen bonds and pi–pi interactions [7]. More importantly, though, glutamine is an amino acid of different functionality compared to lysine (K). The mutant 484K, which appeared in the UK, South Africa (previous beta VOC), and Brazil (previous gamma VOC), cannot form pi–pi interactions, as the side chain of lysine (when protonated) lacks pi electrons. Additionally, the side chain of lysine cannot act as an HB acceptor (-NH_3_^+^ group) in comparison with glutamine (-NH_2_ group). In our previous study, with the use of molecular modeling tools, we demonstrated that the mutation E484Q results in the formation of a more stable ACE2-RBD complex compared to E484K [13]. It is worth noticing that in the same study, it is mentioned that the E484R mutation leads to the formation of the most stable ACE2-RBD complex. The crucial role of arginine in infectivity indicates the potential trend that drives the emergence of new SARS-CoV-2 subvariants: the replacement of non-polar and hydrophobic amino acids with polar and hydrophilic and/or mutations to more hydrophilic residues.

#### 4.3.5. The K417N Mutation (South Africa Variant)

The K417N mutation appeared in the beta VOC and led to a mutant with increased stability of the network between the mutated RBD and ACE2. Lysine contains an ε amino group (-NH_3_^+^) capable of potentially forming salt bridges with negative groups such as carboxylates and/or as a donor in hydrogen bonds. Asparagine contains the amide group (-CONH_2_), potentially participating in hydrogen bonds (donor and acceptor) and pi–pi interactions through the carbonyl group. The latter extra functionality of asparagine may allow a stronger affinity of this amino acid with ACE2 compared to lysine residues. Lysine and asparagine residues’ hydrophilicity indexes are in the same order of magnitude, with the former being slightly more hydrophilic [58]. Interestingly, free energy perturbation calculations for the interaction of the K417N mutated S-protein RBD with both the ACE2 receptor and antibody derived from COVID-19 patients have shown that the S RBD-ACE2 interactions were significantly increased, whereas the antibody interaction dramatically decreased [59,60,61,62].

#### 4.3.6. The P681H Mutation (Nigerian Variant) and the P681R (Indian Variant) in the Vicinity of the S1/S2 Spike Fusion Region

The P681H and P681R mutations occur in the neighboring fusion cleavage site of SARS-CoV-2 at position 685R (Figure 5A). They are globally prevalent and are the result of stronger binding of the 681H and 681R mutants with the ACE2 receptor compared to the P681 original variant. Proline is a rigid amino acid that changes the direction of a peptide sequence with the restricted ability to participate in charge networks as it lacks polar groups in the side chain. On the contrary, histidine is a polyfunctional amphoteric amino acid that can be positively charged (when both nitrogen atoms are protonated) and can potentially form pi–pi interactions with aromatic residues or groups bearing pi electrons. This allows increased avidity of histidine with neighboring negatively charged groups or with aromatic residues and participation in a charge relay system as in angiotensin [63,64] and serine proteases [65]. The P681R mutation is even more transmissible and infective compared to P681H. Arginine is more hydrophilic compared to histidine [58], leading to stronger interactions and more stable networks (Table 2) [5]. Again, the trend of replacing residues with more hydrophilic ones results in mutants of increased stability of the RBD-ACE2 complex.

#### 4.3.7. The D614G (India, South Africa, and Brazil Variants) and D614R Mutations

Another S-protein’s mutation, the D614G, has been seen in the previous beta, gamma, and delta VOCs, allowing the virus to replicate more efficiently in the upper respiratory tract rather than in the lower tract in a mice model [11]. This increased viral load in this anatomical site may allow the virus to spread through sternutation and to cough more easily. However, the D614G variant, although circulating for some time, has not been shown to be more pathological. This may be due to the loose interactions that are expected for glycine, with less binding affinity compared to D614 towards ACE2. Substitution of aspartate by glycine leads to a virus variant with fewer interactions of the neutral non-polar hydrophobic glycine residue with the ACE2 protein compared to the functional polar and charged and the hydrophilic carboxyl group of aspartate residue (Table 2). This mutation leads to a less charged network and reduced binding affinity of variant 614G to ACE2. This may account for less infectivity and less severe symptoms when compared to other mutations, which are strongly stabilized in the surrounding charged environment [11,66]. However, potential mutation to D614R may lead to strong interactions with ACE2 residues and higher infectivity. Arginine is a polar, positively charged hydrophilic amino acid stabilizing network compared to the non-polar neutral hydrophobic glycine residue in the hydropathy ranking properties of the twenty common amino acids (Table 2).

#### 4.3.8. The R346T Mutation in Omicron Subvariants BQ.1 and BA.4.6

Currently, the omicron subvariant BA.5 is dominating globally and has shown substantial immune escape as compared with previous omicron subvariants. As can be seen in Figure 1, the same motif applies to the European variants’ distribution (i.e., BA.5 prevalence). Interestingly, the BQ.1 variant (sublineage of BA.5) is gaining ground and possesses the high potential to become the next dominant variant around the globe [37] (accessed on 24 November 2022). BQ.1 carries critical mutations in S-protein’s antigenic sites, including K444T and N460K. In addition to these mutations, BQ.1.1 bears an additional R346T mutation in a key antigenic site on the S-protein of SARS-CoV-2 [67]. BA.4.6 is a sublineage of BA.4 and has recently increased in prevalence in certain regions currently dominated by BA.5. BA.4.6 carries two additional mutations in the S-protein: R346T and N658S. The ability of BA.4.6 to evade neutralizing antibodies and the role of R346T mutation remains to be determined [2,3,68]. The R346T mutation is also the case for the BQ.1 variant, and it may be involved in the molecular recognition with ACE2 that is necessary for viral infectivity and/or transmissibility. Computational studies regarding the RBD-ACE2 complex stability for these novel omicron subvariants may provide critical insights regarding the characteristics of molecular recognition.

### 4.4. The Role of Tyrosine in Stabilizing Network Systems

#### 4.4.1. The Example of AT1R/ARBs

The multifunctional reactivity of tyrosine (hydroxylate, aromaticity) contributes greatly to the creation of stable networks in tyrosine variants. This network is well depicted in ARBs/angiotensin type I receptor (AT1R) crystal interaction, where tetrazolate and carboxylate groups of ARBs, as in olmesartan interact with residues R167 and Tyr35 of AT1R [69,70]. This is also well depicted in AngII/angiotensin type 1 receptor (AT1R) crystal interaction, where mutation Y35A destabilizes the network, resulting in its inactivation and preventing binding of AngII and ARBs with AT1R [69,70]. Alanine is a neutral amino acid that lacks functionality and therefore binds loosely with neighboring residues. In our studies in the renin–angiotensin system (RAS), it has been found that angiotensin II exerts its agonist activity through the tyrosine hydroxylate formed in a charge relay system (CRS) mechanism analogous to the serine protease cleavage mechanism through a serinate anion [63,65]. The interaction of angiotensin II tyrosinate with its AT1 receptor is blocked with angiotensin II receptor antagonists (ARBs), protecting from hypertension and related cardiovascular diseases. Mutation Y35A in AT1R results in an inert biological network and confirms the intriguing central role of Tyr to create and stabilize networks upon the binding of AngII or ARBs with AT1R. In silico studies have demonstrated a strong affinity of ARBs with AT1R and with the S-protein of SARS-CoV-2, in particular with arginine-rich sites as 681–686 and 814–815. Angiotensin II receptor blockers (ARBs) have been found to upregulate ACE2 and to be protective in hypertensive patients infected by SARS-CoV-2, rendering them potential inhibitors that may prevent infection [71]. Agents which upregulate ACE2, such as Diminazene [72,73,74] and angiotensin II inhibitors, reduce the RAS toxic angiotensin II implicated in the storm of cytokines and in pneumonia, one of the symptoms of COVID-19. ARBs may be another promising class of repurposable drugs for inhibiting SARS-CoV-2 [13,14]. Figure 7 shows the intermolecular interactions between AT1R and ARB Olmesartan (PDB 4ZUD).

#### 4.4.2. The Role of Tyrosine in Triggering the Hypertensive Activity of Angiotensin II and Possibly the Proinflammatory Cytokine Storm in COVID-19

Recent studies directly implicate the cytokine storm in COVID-19 patients with over-expression of AngII in the renin-angiotensin system (RAS). An important beneficial function of ACE2 is the degradation of toxic Ang II to beneficial Ang heptapeptides after the decarboxylation of alamandine. This function is a pivotal link between ACE2 deficiency and SARS-CoV-2 infection [75]. Our previous studies on AngII (mechanism of action and rational design of ARBs) have revealed that Ang acts at the AT1R through a charge relay system (CRS), analogous to serine proteases involving residues tyrosine, histidine, phenylalanine, and C-terminal carboxylate [63,65]. Tyrosine is the principal component of RAS, and the tyrosine hydroxylate anion binds to the AngII receptor to elicit a vasoconstrictive effect. Methylation of the tyrosine hydroxyl group eliminates activity revealing the importance of a tyrosinate negative charge for potency [63,64]. The charge relay system creates a cyclic structure within the AngII molecule which at the receptor level operates through the tyrosine hydroxylate to trigger activity [63]. In the interface of ACE2/SARS-CoV-2 interactions, tyrosine forms strong salt bridges with negative groups such as carboxylates (aspartic acid and glutamic acid). Tyrosinates strongly bind to SARS-CoV-2 positively charged residues, such as arginine, to create stable networks with subsequent increased infectivity and morbidity. The structure of the SARS-CoV-2 RBD bound to the ACE2 receptor, and the interactive residues have been reported by Lan and colleagues [12].

#### 4.4.3. pi–pi Interactions Enhance Binding Affinity

Our previous MD simulation results [13] underlined that pi–pi interactions of the two tyrosine residues (i.e., Y41 of ACE2 and 501Y) significantly enhance the binding between RDB and ACE2. The single pi–pi interaction may overwhelm weak HBs in terms of the interaction potential. Additionally, the proximity association between these two residues is most likely due to the pi–pi interaction and not to the weak HB that is probably formed between them, although the latter could indeed make a partial contribution to the molecular recognition of RBD with ACE2. The pi–pi interaction can potentially adopt slightly different conformations, such as T-shaped, sandwich, and parallel-displaced (Figure 8). A computational QM analysis of this phenomenon could be undertaken using isolated tyrosine residues or phenol molecules in vacuo, coordinated or non-coordinated [4,15,16,17]. Our studies indicate that fully atomistic MD simulations can also provide useful insights regarding the intermolecular interactions that dominate the molecular recognition of RBD with ACE2 [13,14].

#### 4.4.4. RAS and ACE2 Are Targets for COVID-19 Antiviral Drugs

The RAS has been the prime target for the therapy of cardiovascular diseases, and non-peptide angiotensin AT1 receptor blockers (ARBs) have been developed to specifically block the AT1 receptor [63]. Since angiotensin-converting enzyme 2 (ACE2) in the RAS is the entry of SARS-CoV-2 in the cell initiating infection, ARBs and bisartans, and in particular BV6 (bearing two biphenyl tetrazoles), were investigated as possible antivirals to treat COVID-19 disease furthermore to its antihypertensive potential [13,14]. Extensive clinical studies have shown that ACE2 and ARBs are beneficial in the treatment of hypertensive patients infected by COVID-19 [76,77]. Other studies looking at the mechanism of triggering disease have shown that imbalance in RAS in favor of angiotensin II deregulates the system, exaggerates SARS-CoV-2 specific T-cells, and contributes to COVID-19 severity and mortality [78,79]. A symptom of morbidity is the release of inflammatory cytokines, and ARBs could be a promising strategy not only for COVID-19 but also for autoimmune diseases. ARBs modulate TH1- and TH17-mediated potency by converting pathogenic cytokines to regulatory [80,81,82]. The conformation of angiotensin, the principal component of RAS, which led to non-peptide mimetic ARBs and Bisartans was the result of pioneer work based on structure–activity studies, nuclear magnetic resonance, fluorescence, and modeling techniques [83,84,85,86,87]. In all these studies, the arginine residues play a catalytic role in basic cleavage sites (R685-S686 and R815-S816) for the S-protein’s cleavage and triggering infection induced by furin and TMPRSS2 proteases [88,89,90]. Furthermore, arginine (L452R) and RBD mutations enhance the binding of RBD S-protein with ACE2 increasing transmissibility and infectivity [5]. Arginine blockers, thus, are potential therapeutics for treating COVID-19 [13,14]. Proteases such as furin, trypsin, TMPRSS2, and 3CL^pro^ are potential targets for designing novel COVID-19 drugs [91,92].

## 5. Conclusions

In this commentary, we report the epidemiology of SARS-CoV-2 variants across European countries and the distribution of the variants. We also focus on the interface of the complex of the S-protein’s RBD and the ACE2, where the intermolecular interactions are the driving force that stabilizes the mutated variants of SARS-CoV-2. Mutations lead to stronger charged networks, which are energetically favored through interactions such as hydrogen bonds, electrostatic, salt bridges, hydrophobic (e.g., pi–pi interactions), and van der Waals. The functionality of the interacting residues and their ability to create stable networks is the key to driving mutations to new pathologically significant alterations in COVID-19 symptoms. The polarity and charges of the mutated residues are the cornerstones directing the pathogenicity of SARS-CoV-2. MD simulations show that the RBD triple mutant (N501Y + E484K + K417N) binds more strongly to the ACE2 receptor [13]. The total energy calculations revealed that the wild-type complex was the weakest, the triple mutant was the strongest, and every single mutant complex for E484K, K417N, and N501Y possessed intermediate stability (i.e., between the wild-type and triple-mutant). The pi–pi interaction between 501Y (RBD) and Y41 (ACE2) dominates the interface’s intermolecular interactions and is the closest atom interaction (~1.6 Å) of all interactions detected after the MD simulation analysis. These driving forces may translate into stronger RBD-ACE2 binding with subsequent higher transmissibility and infectivity. Similarly, the triple mutation L452R, E484Q, and P681H, which appeared in the Indian variant, results in a higher affinity of the S-protein’s RBD for ACE2. Intermolecular and intramolecular interactions stabilize the highly infective SARS-CoV-2 mutations N501Y, E484K, E484R, K417N, P681H, and P681R on the S-protein and strengthen the complex stability of RBD-ACE2. These interactions may render novel treatment targets by disrupting the RBD-ACE2 binding with inhibitors and thus preventing infection by SARS-CoV-2. The P681R mutation at the rich arginine FCS 680–686 (SPRRARS) enhances the cleavage and the subsequent infectivity suggesting a critical role for basic arginine residues in the transmissivity of SARS-CoV-2. The trend of mutations is, in general, the replacement of non-polar and hydrophobic amino acids with polar and hydrophilic amino acids, which can potentially create more stable networks allowing the mutant to escape the immune system, increasingly spreading. Our studies suggest that arginine blockers such as ARBs can be a class of potential drugs for treating COVID-19.

## Figures and Tables

**Figure 1 viruses-15-00309-f001:**
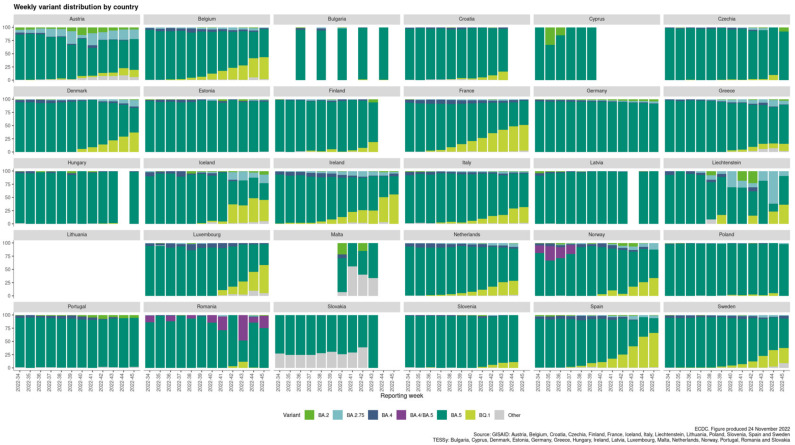
Weekly variant distribution plot by country [19] (accessed on 24 November 2022).

**Figure 2 viruses-15-00309-f002:**
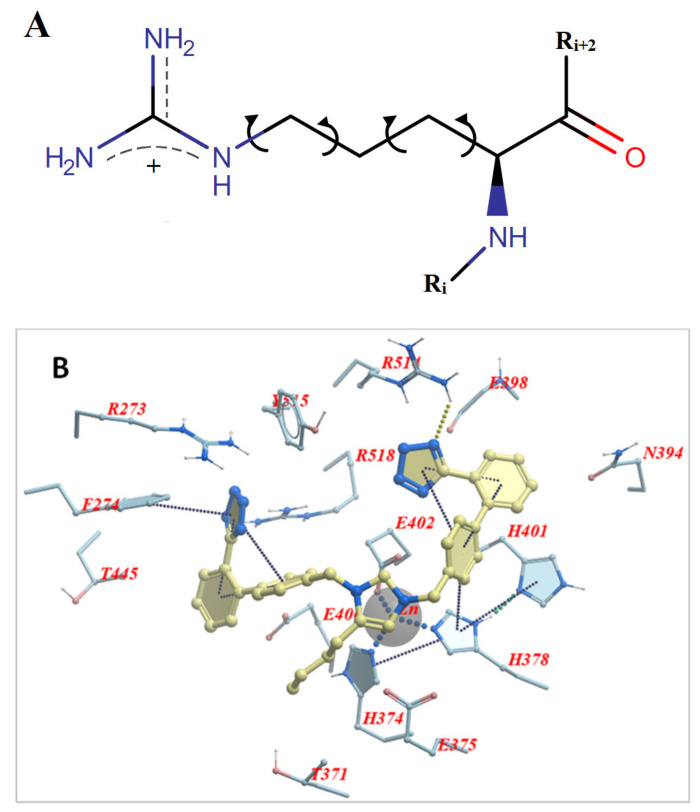
(**A**) Chemical structure of L-arginine in which the torsion angle variables of the side chain are indicated with arrows. The cationic guanidinium moiety forms the dominant part of the arginine side chain. (**B**) Docking pose of bisartan A (4-butyl imidazole bearing two N,N′ biphenyl tetrazole groups) in the ACE2 receptor (PDB 6LZG) and the interaction of Arg518 and Arg514 with one tetrazole of the ligand. The other tetrazole interacts with Arg273 and Arg518. (**C**) Chemical structure of BisA. Note that both tetrazoles are ionized at physiological pH yielding a net charge of −1 for the ligand (illustrations made with MarvinSketch [41]).

**Figure 3 viruses-15-00309-f003:**
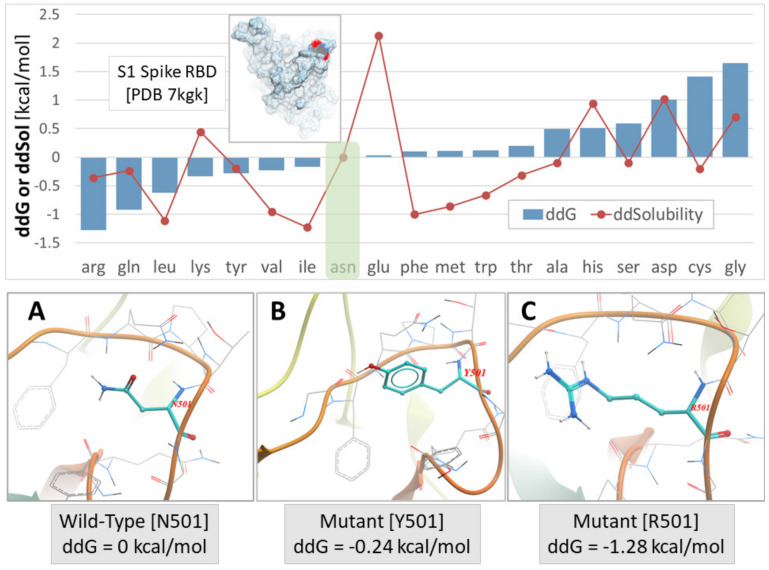
Upper Panel: S-protein stability expressed by change in total free energy (ddG) or free energy of solubility (ddSol) as a function of mutation type (amino acid substitution) at the 501 locus of the PDB ID 7KGK. Lower values of ddG and ddSol correspond to increased protein stability. Thus, an asparagine to arginine substitution at the x501 locus resulted in the largest enhancement of protein stability (−1.28 kcal/mol). Green shaded box indicates the wild-type N501 S-protein. Lower Panel: (**A**) Asparagine (Asn) conformation at the wild-type N501 locus. (**B**) Mutant Y501 conformation. (**C**) Mutant R501 conformation (adapted from [13]).

**Figure 4 viruses-15-00309-f004:**
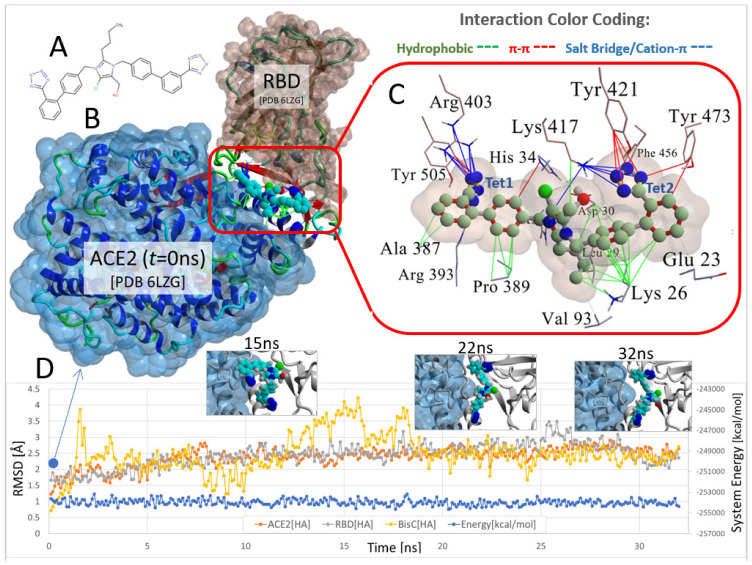
Docking and MD simulation of the ACE2-RBD complex (PDB 6LZG) with the bound imidazole biphenyl-tetrazole (bisartan) BisC. (**A**) Chemical structure of BisC is depicted in the ionized tetrazole form that predominates at neutral pH. (**B**) The ACE2-RBD-BisC complex following BisC global docking at the ACE2-RBD interfacial domain (see docking protocol in Materials and Methods). This docked complex is also the initial starting conformation for the MD simulation. (**C**) Details of the main intermolecular interactions involved in BisC docking at the ACE2/RBD complex. The principal stabilizing forces involved salt-bridge (also referred to as cation–pi) type interactions between the anionic tetrazole group-1 (Tet1) and Arg393 and Arg403 (blue lines). The anionic tetrazole group-2 (Tet2) interacted with Lys417. Additional pi–pi stacking (red lines) and hydrophobic (green lines) interactions also stabilized BisC in the interfacial domain. (**D**) Heavy atom (HA) RMSD trends for the ACE2 receptor (orange), bound RBD (gray), and bound BisC (yellow) over a 32 ns MD simulation of the ACE2-RBD-BisC complex at 311 K and constant pressure of 1 atm (NPT ensemble) in physiological saline (0.9 wt% NaCl solution; refer to Material and Methods for the detailed MD protocol). Inset images show BisC conformations at 15 ns, 22 ns, and 32 ns. While BisC was relatively stable over the course of the MD simulation, it was not ejected into the aqueous phase. Brief periods of sporadic instability were noted (e.g., ~12–17 ns).

**Figure 5 viruses-15-00309-f005:**
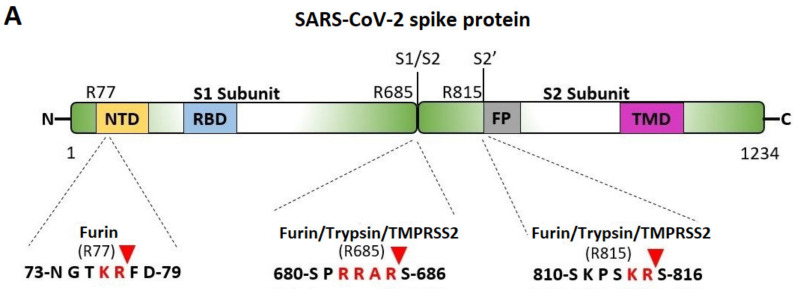
(**A**) Cleavage sites of the SARS-CoV-2 virus S-protein; NTD, N-terminal domain; RBD, receptor-binding domain; FP, fusion peptide; TMD, transmembrane domain. (**B**) Homotrimeric SARS-CoV-2 S-protein and locations of the S1/S2 and S2′ FCSs.

**Figure 6 viruses-15-00309-f006:**
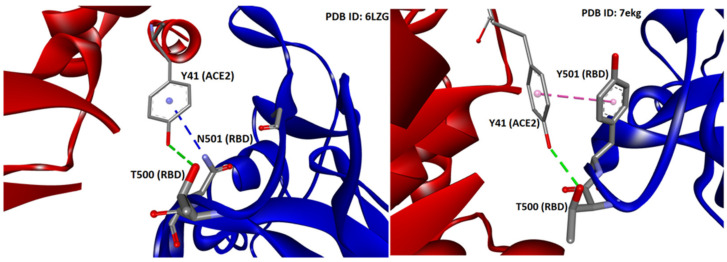
Interactions between Y41 of ACE2 with N501 (left panel) and Y501 (right panel) of S-protein of SARS-CoV-2 (PDB IDs: 6LZG and 7EKG, respectively). HBs are depicted in green (conventional) and blue (Pi-donor HB) and hydrophobic interactions in magenta (Pi–Pi T-shaped). Interactions were analyzed and illustrations made with Discovery Studio Visualizer [56].

**Figure 7 viruses-15-00309-f007:**
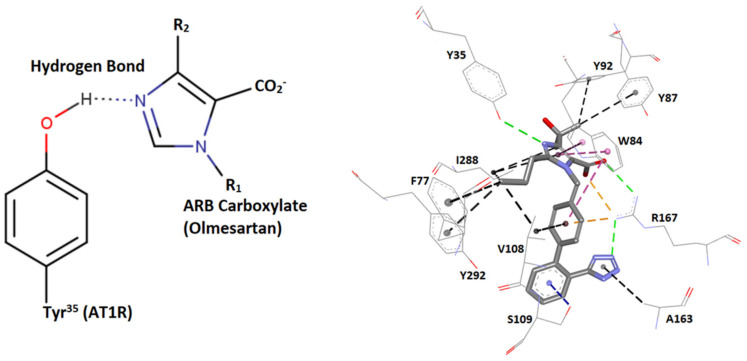
Intermolecular interactions between AT1R and the ARB olmesartan (illustrations made with MarvinSketch 22-11 [41]) and Discovery Studio Visualizer [56]. Left: hydrogen bond interaction between AT1R Y35 hydroxyl group with olmesartan imidazole nitrogen. Right: interactions of the AT1R residues Y35, F77, W84, Y87, Y92, V108, S109, A163, R167, and I288 with the ARB olmesartan (PDB 4ZUD). HBs are depicted in green (conventional) and blue (Pi-donor HB), hydrophobic interactions in black (alkyl-alkyl and Pi-alkyl) and magenta (Pi–Pi stacked and Pi–Pi T-shaped), and electrostatic (salt bridges and Pi–cation) in orange. The Y35A mutation destabilizes the Olmesartan/AT1 network and prevents binding. Note that ARBs may be inhibitors of SARS-CoV-2 and ACE2 binding through strong interactions between negatively charged tetrazolate and carboxylate with polybasic arginine residues’ cavity loop 681–686 RBD of SARS-CoV-2.

**Figure 8 viruses-15-00309-f008:**
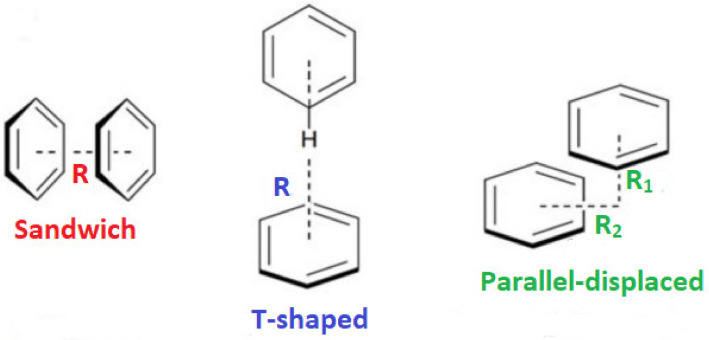
Sandwich, T-shaped, and parallel-displaced configurations of the benzene dimer.

**Table 2 viruses-15-00309-t002:** Physicochemical classes and properties of the 20 common amino acids [58].

Volume Classes	“Hydropathy “ Classes									
	Hydrophobic			Neutral			Hydrophilic			
very large	F		W		Y					
large	I	L	M				K	R		
medium						H				
small	V		C	P	T				E	Q
very small	A			G	S				D	N
	aliphatic		sulfur		hydroxyl	basic			acidic	amide
					uncharged	charged				uncharged
	Non-polar				Polar					

## Data Availability

ECDC country overview report for the 46th week of 2022 is available online on https://www.ecdc.europa.eu/en/covid-19/country-overviews (accessed on 24 November 2022).

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
