# Peer review of "Molecular Epidemiology of SARS-CoV-2: The Dominant Role of Arginine in Mutations and Infectivity"

_viruses, 2023, doi:10.3390/v15020309_

Round 1

Reviewer 1 Report

the molecular epidemiology distribution of SARS-CoV-2 variants across European countries and variant prevalence and the driving forces that trigger mutations in SARS-CoV-2, practically arginine. Generally, the paper is well written and logic in this paper is clear. This rounded analysis can lay a foundation for developing potential drugs for treating Covid-19. Figure 1 could be improved clearer for publishing.

Author Response

Reviewer 1.

We would like to thank Reviewer 1 for his positive comments for our paper. As suggested Figure 1 is improved in analysis. Also the legend is now described in detail. Paper has been reorganized, edited and considerably improved based on comments of reviewers.

Reviewer 2 Report

Dear Authors,

I congratulate all the Authors for their contributions to the writing of the manuscript entitled “Molecular epidemiology of SARS-CoV-2: The dominant role of arginine in mutations and infectivity”.

I have several comments on the manuscript:

1.    Line 36, “by us and others” can be omitted.

2.    Line 40, what is “3CLpro”? Spell it out.

3.    Abstract – the aim, methods and results of the study were not properly described.

4.    Lines 61-64, reference?

5.    Lines 64-66, reference?

6.    Lines 67-69, reference?

7.    Lines 73-76, reference?

8.    The introduction lacks references. Please cite all references where appropriate. Without references, the introduction is highly speculative.

9.    Methods; what are ECDC? GISAID? TESSY databases? VOC/VOI? Please spell them out.

1    10. Methodology is poorly written. The Authors should properly describe the protocols used in the study so that the results could easily be reproduced. How the docking between RBD and ACE2 was performed? What parameter/grid size was used? How many conformations were generated from the docking?

11.  Lines 102-104, full description and abbreviations of COVID-19 and SARS-CoV-2 should be written when they are first mentioned in the text.

12.  Please mention the Pangolin lineage designations for alpha, beta, gamma and delta variants.

13.  Line 121, full description of VOCs is mentioned earlier. It does not need to be mentioned again.

14.  Lines 124, 130, 138 and 143, please reformat the webpage references as numbers. The Authors could use reference manager softwares such as Endnote or Mendeley to format all references throughout the manuscript.

15.  Line 164, see above comment.

16.  Table 1, n should in italics.

17.  Line 182, is this a new subsection? Please reformat the section title.

18.  Lines 219-223, reference?

19.  The Discussion does not reflect the Results obtained in the study. The Discussion reads a review of different mutations and aspects of SARS-CoV-2 such as multi-basic cleavage sites, triple mutations, the role of tyrosine, and other different mutations but the Results report two findings only: variant distribution and the role of arginine. The Authors should discuss the results obtained and compare them with other published studies.

20.  Since the manuscript contains a lot of interesting factual material, I would suggest the Authors to convert it into a review article, which may be useful to the interested readers.

21.  Overall, the format and flow of the manuscript seem to be poorly written. This possibly due to different authors contributed to the writing of the different parts of the manuscript. I would suggest the Main Author to properly check and reformat the manuscript so that it has a proper structure and flow.

I consider the manuscript is sufficiently comprehensive, however it needs a thorough check for its grammar/use of punctuations and proper editing/structuring according to these comments, especially the discussion section. It does not reflect the title and results obtained in the study. The introduction needs references where appropriate.

My sincere congratulations to all Authors.

Author Response

Reviewer 2. 

We are grateful to Reviewer 2 for his positive attitude and for his very constructive criticism and suggestions, which helped us to improve greatly the quality of the manuscript. The paper has been reorganized and rewritten based on his comments.

In particular, see our response to each one of his comments (please be so kind to note that some LINES may differ from the initial manuscript).

  1. Line 36, “by us and others” can be omitted.

Reply:

Phrase “by us and others” is deleted, as suggested.

  1. Line 40, what is “3CLpro”? Spell it out.

Reply:

3CLpro is spelled out, as suggested. (3-chymothrypsin-like protease)

  1. Abstract – the aim, methods and results of the study were not properly described.

Reply: 

As suggested Abstract is now properly described as follows:

Background, Aims, Methods, Results, Conclusion. 

4, 5, 6, 7, 8 (lack of proper references)

Reply: 

As suggested the Introduction references are now cited to support the content and message of article.

  1. Methods; what are ECDC? GISAID? TESSY databases? VOC/VOI? Please spell them out.

Reply:

All abbreviations (i.e., ECDC, TESSY, VOC/ VOI, etc.) are now spelled out previously in Abstract and Introduction. 

  1. Methodology is poorly written. The Authors should properly describe the protocols used in the study so that the results could easily be reproduced. How the docking between RBD and ACE2 was performed? What parameter/grid size was used? How many conformations were generated from the docking?

Reply:

Methodology is now detailed reflecting the findings that are present in the Results section. Details of dockings are described.

  1. Lines 102-104, full description and abbreviations of COVID-19 and SARS-CoV-2 should be written when they are first mentioned in the text.

Reply:

Full description and abbreviations of Covid 19 and SARS CoV 2 are now written.

  1. Please mention the Pangolin lineage designations for alpha, beta, gamma and delta variants.

Reply:

The lineage designations for all SARS CoV 2 variants are now mentioned.

  1. Line 121, full description of VOCs is mentioned earlier. It does not need to be mentioned again.

Reply:

Description of VOCs is now not repeated after the initial description.

  1. Lines 124, 130, 138 and 143, please reformat the webpage references as numbers. The Authors could use reference manager softwares such as Endnote or Mendeley to format all references throughout the manuscript.

Reply:

All webpage references throughout the manuscript are now reformatted as numbers as suggested.

  1. Line 164, see above comment.

Reply:

The same applies as for comment 14.

  1. Table 1, n should in italics.

Reply:

Table 1, n is now in italics 

  1. Line 182, is this a new subsection? Please reformat the section title.

Reply:

Yes, this is a new subsection 3.2 in Results.

  1. Lines 219-223, reference?

Reply:

These lines are referenced 

  1. The Discussion does not reflect the Results obtained in the study. The Discussion reads a review of different mutations and aspects of SARS-CoV-2 such as multi-basic cleavage sites, triple mutations, the role of tyrosine, and other different mutations but the Results report two findings only: variant distribution and the role of arginine. The Authors should discuss the results obtained and compare them with other published studies.

Reply:

Variant distribution and the critical role of arginine in mutations and infectivity are separately discussed in the beginning of Discussion section in separate subsections. 

  1. Since the manuscript contains a lot of interesting factual material, I would suggest the Authors to convert it into a review article, which may be useful to the interested readers.

Reply:

The revised manuscript contains new docking and Molecular dynamics (MD) results, and thus we believe this should be published as article. 

  1. Overall, the format and flow of the manuscript seem to be poorly written. This possibly due to different authors contributed to the writing of the different parts of the manuscript. I would suggest the Main Author to properly check and reformat the manuscript so that it has a proper structure and flow.

Reply:

The revised manuscript has now a proper structure and flow. Title, Abstract, Introduction, Results and Discussion are in line. Grammar is also checked. 

We believe the current revised article in this shape is comprehensive, reads well and is a significant contribution to the field and science.

We thank all reviewers for their useful comments.

Round 2

Reviewer 2 Report

Again, I congratulate all the Authors for their contributions to the writing of the manuscript.

I believe that the manuscript is now suitable for publication in the Viruses journal and can be accepted after minor spell checking/editing.